# Membrane-Based Solutions for the Polish Coal Mining Industry

**DOI:** 10.3390/membranes11080638

**Published:** 2021-08-18

**Authors:** Krzysztof Mitko, Marian Turek

**Affiliations:** Faculty of Chemistry, Silesian University of Technology, ul. B. Krzywoustego 6, 44-100 Gliwice, Poland; marian.turek@polsl.pl

**Keywords:** industrial effluents, coal mine waters, desalination

## Abstract

Poland still relies largely on coal for energy generation, which creates environmental problems connected to the mining process, particularly the issue of saline waste water discharge. Membrane-based zero liquid discharge systems can be employed to recover important resources from coal mine waste waters, such as demineralized water, salt, magnesium hydroxide, and gypsum. In this paper, we present a historical overview of proposed membrane-based solutions for the Polish coal mining industry and discuss possible further areas of research.

## 1. Introduction

One of the industries that may benefit from the implementation of membrane-based technologies is coal mining. Despite the fact that coal mining is being phased out in favor of alternative, cleaner sources of energy, its environmental effects will not disappear any time soon. Moreover, coal is still very important for the metallurgical industry, and for this reason, coking coal is considered to be a critical raw material by the European Union [1]. The European metallurgy industry relies on domestic coking to meet 37% of its needs and accounts for 95% of coke usage [2]. Poland produces 95% of hard coal in the EU [3].

Depending on the geology of the site, coal can be excavated by surface mining (about 40% of worldwide coal production) or underground mining. Coal excavation generates various types of emissions to the environment: solid waste (coal refuse), air emissions (e.g., methane), and waste waters. Underground mining in particular generates a lot of waste water (about 3 m^3^ per 1 t of coal) [4]. Depending on how deep the coal seam is, the salinity can vary from low-salinity brackish water to brine with a salinity 2 to 3 times higher than that of sea water. Coal mining waste waters, rich in salts and minerals, could be a viable feed for resource recovery using membrane-based technologies. The utilization of saline effluents, generated due to the necessity of dewatering both active and closed mines, is a major problem, especially for inland mines where discharge to the sea is not an option. The latter is specifically the case in Poland.

To our best knowledge, there is no comprehensive review of the technologies proposed for the Polish coal mining industry. Thus, in this paper we discuss the current state of affairs in the Polish coal mining industry, give a historical perspective of the development of membrane-based technology in this branch of industry, and discuss the prospects for further development in the future. The review of the approaches taken so far and possible directions for future research should help in setting the goals for researchers and give the industry a general idea of what can be done. The EU’s Water Framework Directive [5] obliges Poland to reach a “good status” of rivers by no later than 2027, which puts pressure on the industry to fix saline waste water problems. Because of that, membrane-based desalination technologies are expected to be an increasingly important field of research.

## 2. Coal Mining Industry

The documented anticipated economic resources in hard coal deposits in Poland as of 31 December 2018 are 61,436 million metric tons. Steam coals constitute 69.6% of the resources, coking coals 29.1%, and other types of coal constitute 1.3% of all coal resources [6]. Because of the large domestic reserves, power generation relies on this raw material. In 50.53% of power plants hard coal is used, and 19.29% use brown coal [7]. The Upper Silesian Coal Basin (UCSB) is the main coal basin of Poland. The area of the UCSB is estimated at about 5600 km^2^. Currently, 80.33% of the documented resources of Polish hard coal are in the UCSB [8]. Only one active coal mine is located outside the UCSB, and all coal mining activities in the UCSB are underground. The coal mining industry has been in decline since the early 1990s. For comparison, in 1990 the entire coal mining sector employed about 388,000 people, whereas in 2020 the employment was only 83,000. Production dropped from 147 to 62 million tons in the same period. The recent (May 2021) government plan states that UCSB coal mines will start to phase out in 2021, with the last coal mine to be closed by 2049. That would help achieve the EU’s goal of net-zero greenhouse gas emissions by 2050, as declared in the European Green Deal; however, closing mines by 2049 means the environmental problems will persist well into the second half of the 21st century. At this point, however, it is still unclear if the national decarbonization plan will be accepted by the EU and if the transitioning of the energy sector will decrease the coal demand so much that the coal mines will have to close sooner.

One of the emissions generated by coal mining is methane, a greenhouse gas that contributes to climate change. Coal mining accounts for 6% of global methane emissions [7]. Polish coal mines are classified as methane (gas) and non-methane (non-gas) mines. In 2016, 993.76 million m^3^ of methane was emitted by UCSB coal mines, of which 342.08 million m^3^ was captured [9].

Another important emission is saline water. North of the USCB, most mines (working and abandoned) are potentially hydraulically interconnected, either directly or indirectly, by drifts, roadways, boreholes, goaf, or intact coal barriers of limited thickness. The objective of dewatering is to keep the water in the abandoned mine under the level of the “over-spill” connection to the adjacent working mines [10]. Mine dewatering is the main cause of the USCB’s hydrochemical regime disturbances. Mine water discharges have an important influence on surface water quality and quantity in the USCB. Mine waters from these mines are typically discharged into tributaries of the upper Vistula and upper Odra rivers [10], where they affect surface water both quantitatively and qualitatively, especially in small streams where significant changes in hydrological regime are caused by large loads of contaminants in a high quantity of mine discharges [11]. For example, a recent study of the effect of saline discharge on the Bolina River (Vistula tributary) found that salinization supports the occurrence and further spread of alien and invasive species and can lead to a decrease in aquatic biodiversity that favors invasive species over native species [12].

Although acidic mine drainage is a major problem elsewhere in the world, in the case of the UCSB it is not significant. The main issue with coal mine waste waters is their high salinity. When discharged, the increase in surface water salinity can have a significant effect on flora and fauna and limits the water use for consumption, agriculture, and industry. The environmental fees paid by the coal mines are proportional to the sum of Cl^−^ and SO_4_^2−^.

## 3. Coal Mine Water Utilization

The proposed methods for solving the saline waste waters problem include controlled drainage of waters (a hydrotechnical method that includes storage in reservoirs and controlled discharge into the rivers when water levels are high), recirculation and forcing the waters into rock mass, and utilization [13].

The utilization of coal mine waters bears the promise of recovering valuable raw materials: demineralized water, sodium chloride, magnesium hydroxide, calcium sulphate, boron, and other chemicals present in the discharge. The recovery of raw materials does not only mitigate the environmental problems caused by the coal mine water discharge, but it can also generate new sources of profit for the coal mining industry. However, the recovery of water alone may not justify the investment in the water treatment technologies. For example, a life cycle analysis of zero liquid discharge systems operating on Chinese coal mine water showed recovered water is not economically competitive, even in regions where the water is scarce (unless the environmental effects are taken into account) [14]. While the conventional methods, such as evaporation with staged extraction of minerals with carnallite [15], can be used for resource extraction, the membrane-based zero liquid discharge (ZLD) systems offer advantages in terms of scalability and energy consumption.

The application of membrane technologies in coal mine water utilization has been studied worldwide. Squires et al. [16] investigated reverse osmosis as a method for desalination of the UK′s coal mine drainage with an average TDS of 1.72 g/L. Reverse osmosis was also studied as a method for treatment of coal mine water with a TDS of 3.959 g/L in South Africa [17]. Chromíková et al. [18] investigated the electrodialytic desalination of effluents originating from Czech coal mines (average salinity of 38.15 g/L as Cl^−^) after pre-treatment with aeration, sedimentation, and filtration; the authors proved high water recovery (up to 99%) can be achieved. A capacitive deionization plant has been running in Shangdong, China [19], on coal mine water with an average TDS of 2.03 g/L. A forward osmosis–reverse osmosis system has been studied using waters from three different Australian mines [20]. Desalination by vacuum membrane distillation has been studied using water with a TDS of 2.3 g/L from Appin, Australia [21]. Analysis of 77 coal mines in Western China has shown that 19.4% of them are using membrane technologies to treat saline waste waters, mainly in areas of high TDS and those near industrial parks [22]. Reverse osmosis was also confirmed to remove major constituents of Xuzhou-Datun coal mine district waste waters, although the trace contaminants were not effectively removed [23]. Reverse osmosis and electrodialysis are increasingly being used in Chinese coal mines [24]. Electrodialysis powered by renewable energy (solar/wind) has been proposed as a method for recovery of demineralized water from evaporation ponds used as a last step in advanced treatment of coal mine water [25]. However, in this paper we focus on the proposals presented for the Polish coal mining industry.

### 3.1. Coal Mine Water Utilization in Poland

#### 3.1.1. The ZOD Technology

Currently in Poland only one desalination plant utilizes saline mine waters for evaporated salt production, with an annual capacity of 100,000 tones [26]. It is located at the Czerwionka-Leszczyny and formerly known as “Dębieńsko” (Figure 1). The feed water for the plant comes from “Budryk” mine, which is still in operation. The plant utilizes the technology known as ZOD, which is based on membrane and thermal methods. Chemical treatment is not used. Two kinds of “Budryk” waters are used: an intermediate one (TDS of 30.8 kg/m^3^) and a brine one (TDS of 72.7 kg/m^3^). The intermediate “Budryk” water is pre-concentrated by reverse osmosis (RO). The RO retentate is mixed with the “Budryk” brine and fed to a vapor compression (VC) evaporator, where it is concentrated to approximately 290 g/dm^3^ as NaCl. The energy consumption of VC is 44 kWh/m^3^ of condensate. The brine is directed to the VC crystallization evaporator, which has an energy consumption of 66 kWh/m^3^ of condensate. The application of integrated membrane systems could decrease plant energy consumption and increase salt recovery [27].

#### 3.1.2. The NF–ED–RO System

Turek et al. [27,28] have proposed an improvement of ZOD technology by the application of a hybrid, membrane-based system (Figure 2). In the proposed technology, the feed water is subjected to high-recovery nanofiltration (NF), followed by electrodialysis (ED) and reverse osmosis (RO). The NF retentate is used to precipitate magnesium hydroxide with sodium hydroxide, while the gypsum precipitates spontaneously from the supersaturated solution. The NF permeate is concentrated by ED. ED concentrate is of sufficient concentration to be used for salt recovery by crystallization, whereas the ED diluate is subjected to RO, producing demineralized water. The RO retentate and the post-crystallization lyes from magnesium hydroxide and gypsum recovery are both recycled to NF. It was estimated that the proposed system would decrease the energy consumption by up to 62%, while simultaneously increasing the salt recovery and producing magnesium hydroxide, which is currently not being recovered with ZOD technology. The proposed high-recovery nanofiltration is currently being tested in a pilot plant located on the premises of the “Dębieńsko” plant in Czerwionka-Leszczyny [29].

#### 3.1.3. The RO–NF System

Another proposed improvement of ZOD technology included the application of NF on the RO retentate [30] (Figure 3). As described above, RO is used in ZOD technology to pre-concentrate the low-salinity coal mine waters before the vapor compression evaporator. Because the RO membranes show similar rejection coefficients to monovalent and multivalent ions, all the magnesium and calcium end up in the RO retentate and eventually in the ZOD crystallizer, where they limit salt recovery. With the proposed improvement, the energy contained by the still-pressurized RO retentate powers the nanofiltration. The NF, in fact, works as an alternative energy recovery device, which also influences the ionic composition. The calculations presented in [30] showed that energy consumption in the RO–NF evaporator system decreased by 40% compared to the RO evaporator system.

#### 3.1.4. The ED–EDR System

The electrodialysis–electrodialysis reversal (ED–EDR) system was also proposed as a replacement for RO in ZOD technology [31] (Figure 4). In the proposed system, coal mine water is subjected to ED, and the diluate is further treated with EDR, which is more scaling-resistant. The EDR concentrate is treated with lime to precipitate magnesium hydroxide and gypsum, and after chemical purification, EDR concentrate is recycled back to the first ED unit. It was found that the lowest energy consumption was achieved by pre-concentrating the coal mine water in the ED–EDR system, then concentrating it further in an evaporator using cheap, low-grade waste heat.

#### 3.1.5. ZERO BRINE Technology

The application of a membrane-based system was tested on a pilot scale in the “Bolesław Śmiały” coal mine [32,33,34]. The tested ZERO BRINE technology included two-pass nanofiltration with intermediate crystallization, a hybrid reverse osmosis-electrodialysis (ED) system working on the NF permeate, and the evaporator working on the ED concentrate (Figure 5). The results confirmed that the application of nanofiltration is very desirable, as it would not be possible to subsequently recover magnesium hydroxide or gypsum without selective removal of bivalent salts from the feed water. The energy consumption of the proposed technology was substantially lower (33%) than that of the reference ZOD technology, reaching 11.2 kWh/m^3^ of treated brine; however, the economic analysis shows the cost of salt production is higher than the market value of sodium chloride [35]. Currently, there is a follow-up project in which the proposed technology is tested on a bigger scale in the “Piast-Ziemowit” coal mine [36]. Micari et al. [37] modelled the integrated system working on coal mine water, including two-pass nanofiltration with intermediate crystallization, reverse osmosis, and membrane distillation/evaporator. They found that the base case scenario is economically competitive, resulting in a salt price of 100 $/t.

#### 3.1.6. Other Research

Electrodialysis and nanofiltration were proposed to be part of a comprehensive zero liquid discharge utilization system of “Wesoła” coal mine water [38]. In the coal mine in question, two kinds of saline effluents exceed the legal limit for discharge into the surface water: 665 m level of concentration of 35.5 g/L as Cl^−^ and 465 m level of concentration of 2.25 g/L as Cl^−^. In the proposed system, ED concentrates the low-salinity water, while the high-salinity water is treated with NF, then concentrated using the twig towers. The obtained brines are mixed and concentrated in a thermal system ending with salt crystallization.

Electrodialysis has also been investigated as a method for coal mine water utilization. A cascade of four electrodialyzers followed by multi-stage flash distillation (MSF) was proposed for the treatment of “Janina” coal mine water [39]. It was found that the ED–MSF desalination plant should be profitable if the ED concentrate has over 90 g/L as Cl^−^, the price of salt would be 30 $/t, and salt recovery of 85% could be achieved. The integrated ED–electrodialysis reversal (EDR) has been proposed either as a pre-treatment to the MSF-crystallization system or as a method of directly producing saturated brine for the crystallizer [40]. The ED–thermal system was also proposed as a treatment for brackish coal mine water with a TDS of about 10 g/L [41]. The comparison of electrodialysis and thermal method for the utilization of coal mine waters with a TDS of 56.3 and 97.0 g/L shows the electromembrane method exhibits a lower energy consumption than the thermal one [42]. The electrodialysis was also estimated to be able to generate saturated brine using “Bolesław Śmiały” coal mine water, provided it would be pre-concentrated by reverse osmosis [43].

Apart from salt and minerals recovery, the removal of radioactive radium isotopes from coal mine water has been studied. The underground radium removal plants have been in operation in two coal mines since 1999 and 2006 [44].

## 4. Prospects for Future Improvement

The main issue in the implementation of membrane-based desalination systems in coal mines is energy consumption. It is very hard for the salt obtained from industrial waste water to compete with mined rock salt, and while the recovery of other minerals, such as magnesium hydroxide, helps to increase the profitability of desalination plants, the industry as a whole is not very keen on investing in new technologies unless it is forced to do so by new environmental regulations.

A decrease in desalination costs can be achieved by utilizing the methane collected at the site to power the desalination plant. In Polish mines, a single shaft emits between 270,000 and 1,400,000 m^3^n/h of an air–methane mixture, usually with a safe methane concentration in ventilation air of less than 0.7%, which corresponds to a power of 18.7 MW–96.9 MW lost to the environment [45]. Methane is a more potent greenhouse gas than carbon dioxide, so there is increased pressure to capture it instead of simply emitting methane into the atmosphere. Methane capture already exists in Polish coal mines; for instance, in 2017, 19,927.9 thousand m^3^ of methane was captured at the Budryk mine, of which 11,825.9 thousand m^3^ (59%) was utilized [46]. Ostrowski et al. [45,47] proposed a thermal-based desalination system, powered either by captured methane or a mixture of captured methane and hard coal, to generate salt and demineralized water out of coal mine water with a salinity of 12.6 g/L. One could expect that applying the reverse osmosis or NF–RO powered by the electricity generated by captured methane might be beneficial to the proposed technology, as pressure-driven membrane methods are generally less energy-consuming than thermal methods in the tested feed salinity range. In 2018, three of the largest Polish coal mining companies—Polish Mining Group (PGG), Jastrzębie Coal Company JSC (JSW), and Tauron Excavation JSC—together with the national gas and oil monopoly, launched Polish Oil Mining and Gas Extraction (PGNiG), the Geo-Metan project, aimed at large-scale demonstration of using methane collected in the “Budryk” and “Bielszowice” coal mines as a source of energy [48]. However, in July 2020, the PGNiG pulled out of the project, citing high costs [49].

Membrane technologies can also be utilized in the methane capture process. For instance, He and Lei [50] investigated the recovery of methane from coal mine ventilation air using inorganic membranes (high-performance silicoaluminophosphate-34 and carbon molecular sieves), finding that, according to techno-economic analysis, such solutions can be feasible. Methane captured in German coal mines and enriched using silicone-based gas separation membranes was tested as a possible fuel for fuel cells [51]. To our best knowledge, however, gas separation membranes were not employed in Polish coal mines for methane enrichment.

An alternative power source for membrane-based desalination technologies could be to use the coal mine waters to produce geothermal energy. The recovery of energy from the coal mines was tested in the Dutch mining industry in the Minewater project [52]. The UCSB waters have temperatures in the range of 13–25 °C, and there are a few installations in the region which utilize this energy for heating and cooling, such as the Central Mine Dewatering Company (CZOK) offices in Czeladź, miners’ bathhouse in the “Silesia” coal mine, or Silesian Museum in Katowice [53]. Geothermal energy has not yet been used in saline waste water utilization.

## 5. Conclusions

Poland’s biggest rivers have been suffering from saline discharges originating from the coal mining industry. Membrane technologies, such as nanofiltration, reverse osmosis, and electrodialysis, may be used in membrane-based zero liquid discharge desalination systems, aimed to mitigate ecological issues and recover and recycle as much raw material as possible. Gas separation membranes can be also employed in recovery of methane and its subsequent usage as a fuel.

## Figures and Tables

**Figure 1 membranes-11-00638-f001:**
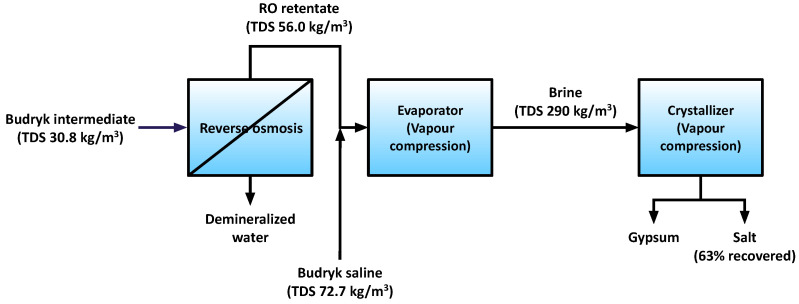
An overview of the ZOD technology for coal mine water treatment, made with data from [27].

**Figure 2 membranes-11-00638-f002:**
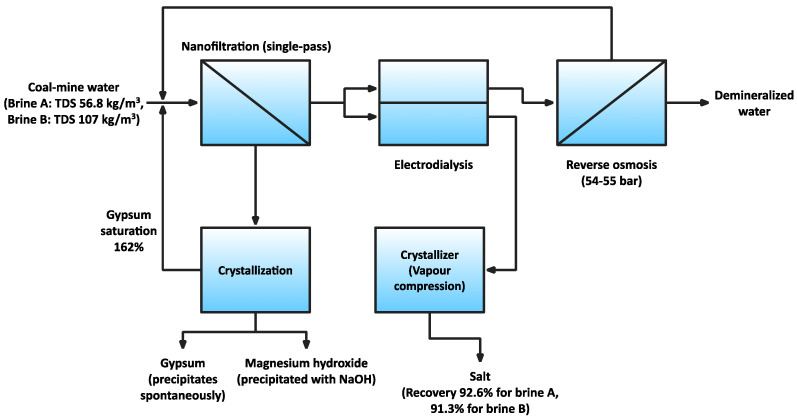
An overview of the proposed NF–ED–RO system, made with data from [27].

**Figure 3 membranes-11-00638-f003:**
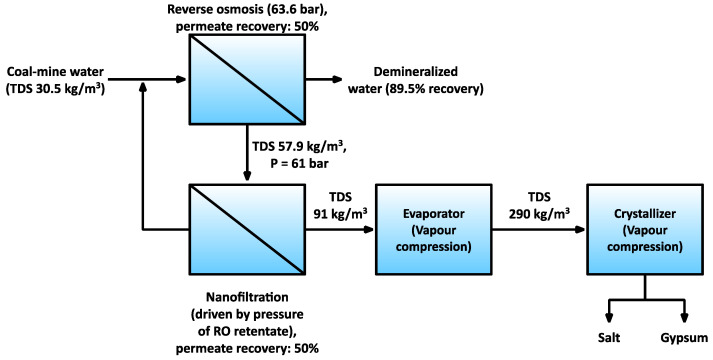
An overview of the proposed RO–NF system [30].

**Figure 4 membranes-11-00638-f004:**
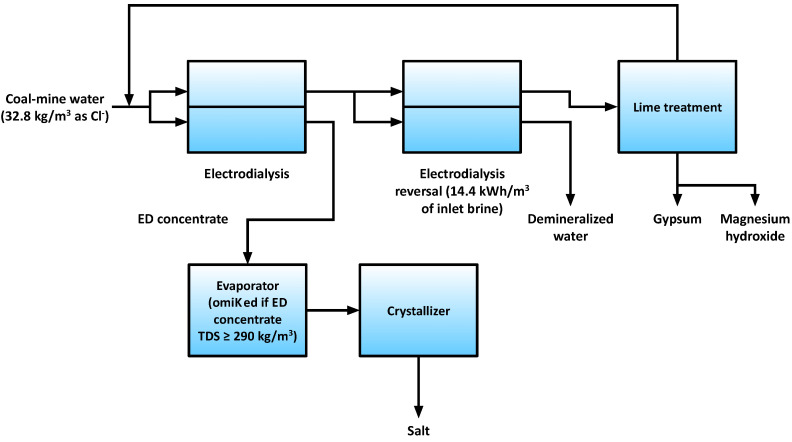
An overview of the proposed ED-EDR system [31].

**Figure 5 membranes-11-00638-f005:**
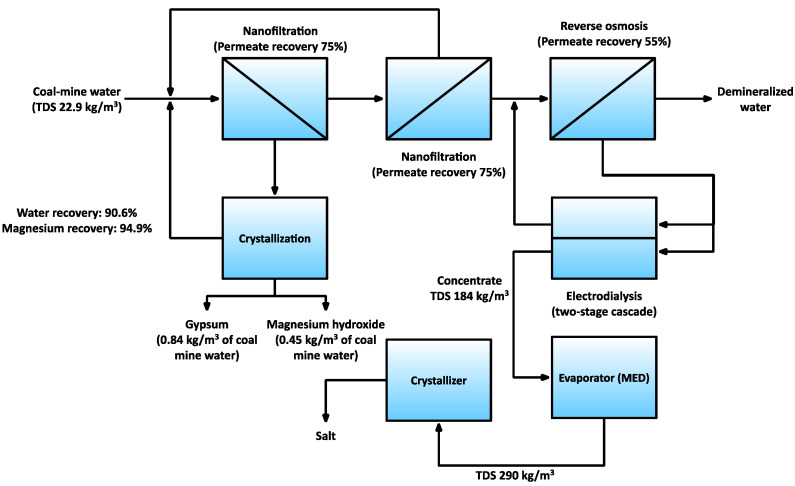
An overview of the ZERO BRINE technology for coal mine water treatment [32].

## Data Availability

Data sharing not applicable.

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
