# Peer review of "Membrane-Based Solutions for the Polish Coal Mining Industry"

_membranes, 2021, doi:10.3390/membranes11080638_

Round 1
Reviewer 1 Report
1) I believe this is an important matter (eg there is always research about acid mine drainage) but I havent heard about salinity in mine waters however the way it is presented is completely underused. you state Although the acidic mine drainage is a major problem elsewhere in the world, in the 51
case of UCSB it is not significant - the main issue with coal mine waste waters is their high 52
salinity.
you mean that this is a problem only in this area? is this also a problem in other parts of the world?
2)as before-in the introduction you have to show why this is important for an international audience
3) I understand how important coal still is however you have to put this situation into context in relation to the European Green Deal, the phasing out of non renewable energy, the need for restoration of mines, the time schedule of phasing out of coal in Poland etc..
4) Fig 1 should better be a google earth or some other similar map
5) Fig 2 to 5-are these Figures takend from the actual papers? you know this is not acceptable. If these Figures are made by you then you should note that "with data from..." also Figures are quite simplistic-is there any way that they can be more informative? they all look the same
6) since this is a short and concise review please make more distinct the different solutions that are proposed in this paper eg make subsections for each part of the discussion
Author Response
Thank you for the review. We believe we fixed the issues you pointed out - the changes are marked in yellow in the revised version of the manuscript.
“1) I believe this is an important matter (eg there is always research about acid mine drainage) but I havent heard about salinity in mine waters however the way it is presented is completely underused. you state Although the acidic mine drainage is a major problem elsewhere in the world, in the 51
case of UCSB it is not significant - the main issue with coal mine waste waters is their high 52
salinity.
you mean that this is a problem only in this area? is this also a problem in other parts of the world?”
The salinity is also a problem in other parts of the world - mostly whenever the underground mining is used (e.g. China and South Africa have the same problems, but not Australia - they use surface mining). We have added references to the research done on the Chinese coal mining industry to show this is not a local problem. The paper focuses on the Poland because that is the topic of the special issue we are submitting to.
“2)as before-in the introduction you have to show why this is important for an international audience”
We have tried to do that by elaborating more on research done in other countries.
“3) I understand how important coal still is however you have to put this situation into context in relation to the European Green Deal, the phasing out of non renewable energy, the need for restoration of mines, the time schedule of phasing out of coal in Poland etc..”
We have added the information on deadline for improving river quality set by the Water Framework Directive (2027) and on the recent goverment plan to close the mines by 2049, which would meet the EU’s goal of climate neutrality by 2050 (although it is still completely unclear how to transition the energy sector, largely reliant on coal). It is very hard to say if this is a final plan (the Brussels only got it in June 2021 and hasn’t given the official response yet - unofficial rumors are they think 2049 is way too long and will be pushing to earlier date of 2035). The goverment also had very limited reaction to the recent (July 14th) “Fit for 55” initiative by the European Commission, which would had to affect phasing out coal mining by changing the emission trading system.
“4) Fig 1 should better be a google earth or some other similar map”
We decided to remove this figure.
“5) Fig 2 to 5-are these Figures takend from the actual papers? you know this is not acceptable. If these Figures are made by you then you should note that "with data from..." also Figures are quite simplistic-is there any way that they can be more informative? they all look the same”
Yes, they were made by us with data taken from the literature. We fixed them to make it clear and to make them more informative.
“6) since this is a short and concise review please make more distinct the different solutions that are proposed in this paper eg make subsections for each part of the discussion”
We have split the discussion into subsections to make it clear.
Reviewer 2 Report
It is a very good study with overall adequate presentation of experimental results. Some additions are needed:
1) Authors should further emphasize on the novelty of their work.
2) Some minor typos, grammar and syntax errors should be carefully revised and corrected accordingly.
3) Reference can be even more updated (more recent relative works).
4) The Introduction is too short. It must be enriched.
5) Be careful about copyright licenses which must be taken if any figure was reproduced from already published paper
Author Response
Thank you for the review. We believe we fixed the issues you pointed out - the changes are marked in yellow in the revised version of the manuscript.
“1) Authors should further emphasize on the novelty of their work.”
We have given more explanation of why we believe this work is new and important:
“To our best knowledge, there is no comprehensive review of the technologies proposed for the Polish coal mining industry. (…) The EU’s Water Framework Directive obliges Poland to reach a “good status” of rivers by no later than 2027, which puts the pressure on the industry to fix the saline waste water problems. Because of that, the membrane-based desalination technologies are expected to be increasingly important field of research.”
“2) Some minor typos, grammar and syntax errors should be carefully revised and corrected accordingly.”
We re-check the manuscript for typos & errors.
“3) Reference can be even more updated (more recent relative works).”
We have added more relevant work published after 2017, namely [9, 12, 14, 22-25, 34-35, 52-53].
“4) The Introduction is too short. It must be enriched.”
In the introduction, we have elaborated more on the coal mining and why we believe this topic is important.
“5) Be careful about copyright licenses which must be taken if any figure was reproduced from already published paper”
We believe we fixed the copyright issues. The figures were not directly copied from published work, but made from scratch using presented data; we made it clear in the text.
Round 2
Reviewer 1 Report
the manuscript is acceptable now
Reviewer 2 Report
All my comments of the initial submission have been correctly replied and included in the
revised manuscript. The quality of this work has been drastically improved after revision and
therefore I recommend its publication as it is.